# Population Genomics of the Critically Endangered Brazilian Merganser

**DOI:** 10.3390/ani13243759

**Published:** 2023-12-06

**Authors:** Davidson P. Campos, Henry Paul Granger-Neto, José E. Santos Júnior, Pierre Faux, Fabrício R. Santos

**Affiliations:** 1Department of Genetics, Ecology and Evolution, Instituto de Ciências Biológicas, Universidade Federal de Minas Gerais, Belo Horizonte 31270-901, MG, Brazil; davidsonpcampos@gmail.com (D.P.C.); hp.granger98@gmail.com (H.P.G.-N.); jrsantos140782@yahoo.com.br (J.E.S.J.); 2GenPhySE, Université de Toulouse, INRAE, ENVT, 31326 Castanet-Tolosan, France; pierre.faux@inrae.fr

**Keywords:** anseriformes, conservation genetics, neotropical bird, Brazilian Cerrado, endangered species, neotropics

## Abstract

**Simple Summary:**

The Brazilian merganser, a critically endangered duck species in South America, was studied using a population genomics approach. This research focused on the genetic diversity of the mergansers in the four remaining wild populations located in Central Brazil. The results showed that there is a low genetic diversity and high levels of inbreeding in individuals across all locations, with a moderate level of genetic differentiation between them. These findings highlight the need for immediate conservation actions to prevent the decline of the Brazilian merganser population and genetic erosion. Genetic monitoring can help implement appropriate in situ and ex situ management strategies to increase the species’ long-term survival in its natural environment.

**Abstract:**

The Brazilian merganser (*Mergus octosetaceus*) is one of the most endangered bird species in South America and comprises less than 250 mature individuals in wild environments. This is a species extremely sensitive to environmental disturbances and restricted to a few “pristine” freshwater habitats in Brazil, and it has been classified as Critically Endangered on the IUCN Red List since 1994. Thus, biological conservation studies are vital to promote adequate management strategies and to avoid the decline of merganser populations. In this context, to understand the evolutionary dynamics and the current genetic diversity of remaining Brazilian merganser populations, we used the “Genotyping by Sequencing” approach to genotype 923 SNPs in 30 individuals from all known areas of occurrence. These populations revealed a low genetic diversity and high inbreeding levels, likely due to the recent population decline associated with habitat loss. Furthermore, it showed a moderate level of genetic differentiation between all populations located in four separated areas of the highly threatened Cerrado biome. The results indicate that urgent actions for the conservation of the species should be accompanied by careful genetic monitoring to allow appropriate in situ and ex situ management to increase the long-term species’ survival in its natural environment.

## 1. Introduction

The number of threatened birds has increased by about 30% in the last twenty years, from 1107 to 1460 species worldwide [1]. The population declines of many bird species were particularly associated with increasing threats, such as hunting, illegal trade, introduction of exotic species, and degradation of habitats [1]. Therefore, bird conservation studies are urgently needed to measure the consequences of population decline and to plan scientific-based management strategies to prevent eventual extinction [2,3,4].

The Brazilian merganser, *Mergus octosetaceus,* Vieillot, 1817, is the only Mergini duck of the Anseriformes order that is found in the Southern Hemisphere [2]. The conservation status of *M. octosetaceus* is a serious concern because the species has been listed as Critically Endangered (C2a(i)) on the IUCN (International Union for Conservation of Nature and Natural Resources) Red List since 1994 [1]. Although the current population size of the species is not clear, it is known that less than 250 mature individuals live in the wild, and its population trend is decreasing [5]. In addition, the remaining Brazilian merganser population is fragmented and heterogeneously distributed in four different locations in Brazil, with between 140 and 200 mature individuals that are resident in the Serra da Canastra National Park and its surroundings [1].

The historical distribution of *M. octosetaceus* comprised three countries—Brazil, Argentina, and Paraguay—and included the hydrographic basins of the São Francisco, Paraná/Paraguay, and Tocantins (Amazon basin) rivers, occurring in several disjointed areas of the Cerrado and Atlantic Forest biomes, always presenting a low population density [6]. The last records of the species in Argentina and Paraguay date from 1993 and 2002, respectively [7,8,9]. The lack of records, despite active searches for the species in the last two decades, indicates that merganser populations in Argentina and Paraguay, and in the Atlantic Forest areas of Brazil, may be extremely rare or have become locally extinct, likely associated with intense human pressure in these places. Therefore, it is assumed that the present distribution of the species is currently limited to Central Brazil, only to the states of Minas Gerais, Goiás, and Tocantins, where populations are being constantly monitored by the Brazilian Action Plan of Conservation (PAN pato-mergulhão-ICMBio).

Although *M. octosetaceus* is mainly found in protected areas (PAs) of the Cerrado biome, such as Chapada dos Veadeiros National Park (Veadeiros), Jalapão State Park (Jalapão), and Serra da Canastra National Park (Canastra) [6,10,11,12,13,14,15,16], the remaining populations of the Brazilian merganser are separated by a mosaic of regions with intense anthropic pressure [2]. In addition to the PAs, recent records in the Alto Paranaíba region (Paranaíba) of the Minas Gerais state revealed a small population occupying unprotected areas of the Cerrado biome that is surrounded by intense farming and mining activities [2].

Most of the research executed with *M. octosetaceus* was concerned with the ecology, reproduction, habitat, and dispersal of the species [11,13,17,18,19]. Although the Brazilian merganser was described by Vieillot in 1817, it is rarely found in collections from Brazil and Argentina, and the first field research performed with the species occurred in the 1950s at the border of Argentina, Paraguay, and Brazil [20]. Furthermore, intensive ecological studies have been conducted since the 1990s in Canastra, located in Southeastern Brazil, where the largest population of the species has been recorded so far [13,14,18,19].

Genetic studies applied to the conservation of endangered species can drive in situ and ex situ preservation and management strategies (e.g., pairing, reintroduction). However, no genomic data have been used so far for conservation purposes for the remaining populations of Brazilian merganser. Our research group published the only four genetic studies on this species [21,22,23,24], showing preliminary data that indicate a low genetic diversity, which is likely related to a recent bottleneck in the remaining populations. The development of new molecular techniques using Massive Parallel Sequencing allowed researchers to obtain a large amount of genetic variation data, covering most of the genome of the studied species [25,26]. Genomic approaches can be applied to populations of endangered species to investigate many conservation issues in greater detail, such as the characterization of inter-population and inter-individual relationships, spatial structure, inbreeding, kinship, and signs of natural selection along the genome [27,28]. To apply genomics in population studies (of multiple individuals), methods were developed to reduce the genomic complexity of data, such as “Genotyping by Sequencing—GBS” [29]. GBS methods have been used to genotype thousands of single-nucleotide polymorphisms (SNPs) scattered around unknown genomes of many individuals of different plants [30,31] and animals [32,33].

In this work, we present the first population genomic survey of the Critically Endangered Brazilian merganser using SNP data generated through GBS methodology applied to all remaining (and known) populations located at Canastra, Paranaíba, Veadeiros, and Jalapão in Brazil. We aim to characterize the genetic diversity between and within populations and to identify possible barriers to gene flow that may establish distinct population groupings in the area of occurrence of the species.

## 2. Material and Methods

### 2.1. Biological Samples

We used 30 Merganser samples (Appendix A) from all known areas of current occurrence of the species: four from Jalapão (full siblings), four from Veadeiros, seven from Paranaíba (four are full siblings), and fifteen from Canastra (Figure 1). Among these, 20 samples were derived from individuals previously collected in the natural areas of occurrence of the species by the collaborators of the Terra Brasilis Institute, Funatura, Cer Vivo, Projeto Mergus da Chapada dos Veadeiros, Naturatins, and ten blood samples were from captive individuals (Zooparque de Itatiba, Itatiba, SP) derived from eggs collected in natural areas by the National Action Plan for Conservation in Brazil (PAN pato-mergulhão). DNA samples were obtained from several types of tissues [23], such as eggs (embryos, shell), feathers found in abandoned nests, and blood from captured animals that were already part of the Tissue Collection of the Centro de Coleções Taxonômicas of Universidade Federal de Minas Gerais, Brazil.

### 2.2. DNA Extraction

The genomic DNA of these samples were extracted according to the phenol-chloroform protocol [34]. The integrity of the samples was verified through agarose gel (2%) electrophoresis, and the quality of the DNA was checked with Nanodrop 2000. Genomic DNA extractions with ratios 260/280 ≥ 1.75 and 260/230 ≥ 1.8 were selected for the GBS approach (below), and the final DNA quantification was evaluated in Qubit^®^ 2.0. The access to genetic resources of Brazilian biodiversity from the Ministry of Environment was registered as a SisGen/Brazil number A324339.

### 2.3. GBS Library Construction and Sequencing

The DNA samples were submitted to a genomic library preparation protocol adapted from Elshire et al. [29] at the Ecomol Genomic Service of the ESALQ-USP (Piracicaba, Brazil). For the digestion of genomic DNA, the *Pst*I enzyme was chosen because it presented satisfactory in vitro test results and in library constructions with many animal species [35,36], including birds [37]. The digestion reaction was performed separately for each sample, using 10 μL of DNA at a concentration of 10 ng/μL, 3 μL of buffer, 1 μL of *Pst*I enzyme (20 U/μL) (New England Biolabs, Ipswich, MA, USA), and 16 μL of water. This solution was incubated at 37 °C for 60 min, and digestion was confirmed using agarose gel (2%). The digested genomic DNA was linked to different barcodes, ranging from 7 to 9 bp, to identify individual samples after sequencing. The fragmented DNA linked to barcodes was lyophilized in a vacuum centrifuge (45 °C for 120 min) and next resuspended in 19 μL of water and ligated to adapters according to the protocol of Elshire et al. [29], adding 1 μL of T4 DNA Ligase enzyme (New England Biolabs) and 5 μL of 10× buffer. This solution was incubated at 22 °C for 120 min, followed by incubation at 65 °C for 30 min to inactivate the enzyme. After ligation, samples were purified with the QIAquick PCR Purification Kit (Qiagen, Valencia, CA, USA) following the manufacturer’s instructions. The purified products were then amplified by inserting primers with sequences complementary to the restriction fragments and adapters, which linked the PCR products to the oligonucleotides that covered the flow cell. The PCR conditions were as follows: 1× (72 °C—5 min; 98 °C—30 s); 18× (98 °C—10 s; 65 °C—30 s; 72 °C—30 s); 1× (72 °C—5 min, 4 °C—10 min). Amplification products were checked on agarose gel (2%) and purified with AMPure XP magnetic beads (Beckman-Coulter, Brea, CA, USA) to remove small DNA fragments. The library was quantified via real-time quantitative PCR (qPCR) and bioanalyzer (Agilent Technologies, Santa Clara, CA, USA) and subsequently sequenced in Illumina HiSeq 2000 (single end with 100 bp reads).

### 2.4. Variant Calling and Filtering

SNPs were called using the Stacks pipeline [38] and then filtered using VCFtools [39]. The following filters were applied to retain variants: (1) being called in all individuals (--max-missing-count 0), (2) having at least two occurrences of the minor allele over the sampling (--mac 2), (3) being a single nucleotide substitution (--remove-indels), (4) being biallelic (--min-alleles 2 –max-alleles 2), and (5) being at Hardy–Weinberg Equilibrium (--hwe 0.05). In addition, for each pre-selected SNP, we computed the minimal depth throughout all individuals and retained only SNPs with minimal depth greater than or equal to 5.

### 2.5. Genetics Statistical Analyses

VCFtools was also used to: (i) calculate the heterozygosity and inbreeding coefficient (F) for each individual; (ii) infer the degree of relationship/kinship between individuals, using the method developed by Manichaikul [40], which calculates the pairwise kinship of all sampled individuals. In this analysis, kinship values between two individuals greater than 0.354 indicate that they may be the same individual or monozygotic twins, kinship values between 0.177 and 0.354 suggest that they are first-degree relatives (full siblings or parents/children), kinship values between 0.0884 and 0.177 indicate second-degree kinship (half-brother or grandparents/grandchildren), and kinship values between 0.0442 and 0.0884 are considered third-degree relatives.

In the R program v4.2.3 (R Core Team, 2020), we used the packages “VCFR” [41], “poppr v2.8.6” [42], “ape” [43], “RcolorBrewer” [44], “ggrepel” [45], “adegenet” [46], “reshape2” [47], and “ggplot2” [48] to perform the following: (i) multivariate analyses represented in two-dimensional graphs using Principal Component Analysis (PCA) to show inter-individual relationships through genetic distance data between individuals from different populations; (ii) Discriminative Principal Component Analysis (DAPC) to identify groups of genetically related individuals; (iii) Compoplot that represents the probability of population association for each sample of the predetermined localities in a bar graph; (iv) a heatmap with pairwise kinship data normalized by Pearson’s correlation matrix, in which the highest relationship level was assigned a value 1 and the lowest relationship level was assigned a value −1.

The phylogenetic reconstruction using Bayesian Inference (BI) among individuals was performed with the Beast program [49] using the SNAPP package [50], which collected the SNPs for each individual that was designated as a separate OTU. The run was executed with a chain length of 200 million, convergence was checked with Tracer v1.7 [51], and the maximum clade credibility tree was calculated using TreeAnnotator v1.10.

The number of possible population groups (k) and their geographic boundaries were estimated using Geneland v3.2.2 [52], using three independent runs with 30 million iterations each, in a range of 1 to 10 possible clusters, and the uncorrelated allele frequency model, which assumes that allele frequencies between populations can be different. The runs were performed with a model of false null alleles and thinning of 300,000, with results visualized after a burn-in of 100 times, where the convergence of each model was evaluated. The number of population groups without geographic correlation was estimated through the STRUCTURE v2.3.4 program [53], using a model that predicts the possibility of gene flow between populations (admixture model), with 1,000,000 MCMC randomizations and a burn-in of 250,000. Population number parameters (K) from 1 to 10 were evaluated with 10 independent runs for each K. The choice of the appropriate K was made with Structure Harvester [54], and the final graph was generated with Clumpak [55]. Measures of differentiation between geographic populations were estimated through pairwise FST analyses, analysis of molecular variance, and Mantel tests that were performed using the Arlequin 3.5 program [56].

## 3. Results

### 3.1. Variant Calling from Sequencing Reads

The sequencing of the GBS library generated 192,590,988 reads, which were initially filtered with the fastp program [57]. In this initial step, we selected reads that presented a phred score above 20, with at least 30% complexity and without polyG and polyA tails. After this initial filtering step, we obtained 190,578,439 reads.

The initial filtered data were submitted to the Stacks pipeline [38] using the “process_radtags” algorithm, where samples were demultiplexed and further submitted to other cleaning filters. Considering the 190,578,439 reads, 2.8% did not show linked barcodes, 0.3% did not show restriction enzyme sites (TGCA), and 1.4% were low-quality reads that were excluded from the following analyses. After this second sequencing cleanup, we retrieved 182,176,706 reads.

The Stacks pipeline was performed using a de novo alignment approach [38], as *Mergus octosetaceus* has no available reference genome. We found 33,535 SNP variants that went through Stacks filtering steps. We then applied more stringent filters (minimal depth ≥5—a rule equivalent to 100% call rate—at least two occurrences of the minor allele, only biallelic SNPs, at Hardy–Weinberg equilibrium), eventually leaving 923 SNPs for analyses. Filtering on minimal depth (over all individuals) was by far the most stringent rule as it filtered out ~98.7% of the initial set. The remaining filtered variants were SNPs that deviated significantly from the Hardy–Weinberg equilibrium. The genotypes of 923 SNPs for all individuals were submitted to different inter-individual and inter-population analyses.

### 3.2. Kinship and Inbreeding Estimates

In the pairwise kinship analysis, we observed that individuals from the same population displayed a high level of genetic relationship. We identified that individuals (full siblings) from Jalapão are the most differentiated in comparison to individuals from all other populations, followed by individuals from Veadeiros and full siblings from Paranaíba. In contrast, we found that the individual PAR180 has at least a third-degree relationship with 16 of the 30 samples, regardless of location, and the individual CAN018 has a significant relationship (second degree) but only with the individual CAN148. We also identified three full siblings (CAN098-CAN099-CAN100) that were supported by fieldwork information in Canastra (Appendix A).

When we visualized the kinship data in a heatmap graphic (Figure 2) ordered by populations, we observed that Canastra and Paranaíba populations showed a closer genetic relationship, where Paranaíba full siblings PAR206, PAR207, PAR208, and PAR209 are the most differentiated individuals from all others between both populations. Furthermore, we can also observe that individuals from Veadeiros and Jalapão present smaller kinship values when compared to individuals from other areas.

Analysis of inbreeding coefficients (F) per individual (Figure 3) revealed a high proportion of inbred individuals (F > 0), with PAR206 presenting the highest inbreeding coefficient (0.36947) (Appendix A). However, some individuals with higher heterozygosity and lower inbreeding levels were also found. For example, PAR180 presented the lowest inbreeding coefficient (−0.32184), and the Veadeiros population presented the highest number of individuals (VEA066, VEA070, VEA071) with negative inbreeding coefficient values (Figure 3).

### 3.3. Population Structure

As the PCA analysis uses a matrix of genetic differences for a two-dimensional representation, full siblings tend to be tightly associated, which may cause a bias in the analysis, making a group separated further from the others. Thus, we have conducted PCA analysis, including only one individual from each of the full sibling groups that we identified in the pairwise kinship analyses and that were confirmed by fieldwork information. Therefore, we excluded individuals JAL202, JAL203 and JAL204, PAR207, PAR208 and PAR209, and CAN098 and CAN100. The resulting PCA graphic with “unrelated” individuals shows us two major clusters, one formed by individuals from Canastra and Paranaíba on the right side of the graphic and another formed by individuals from Veadeiros and Jalapão on the left side of the graphic (Figure 4).

DAPC analysis showed similar results to PCA, separating the same clusters, but the most genetically related individuals are highlighted. We can see that Paranaíba and Canastra individuals form a large group, and they are relatively differentiated from individuals of Veadeiros and Jalapão (Figure 5).

The Compoplot analysis was used to infer the probability of an individual belonging to a given population. The results also showed a close genetic relationship between Paranaíba and Canastra populations, as well as a slight relationship between Veadeiros and Jalapão populations when compared to the Paranaíba + Canastra group (Figure 6A).

To infer the number of population clusters without geographic assignment, we used the Structure program with the uncorrelated allelic frequency model. Two clusters were found, one formed by individuals from Canastra and Paranaíba and another by Jalapão and Veadeiros individuals (Figure 6B).

To characterize independent population groups distributed in a spatial landscape (using geographic coordinates of the samples), we used Geneland analysis with an uncorrelated allelic frequency model. Three population groups were identified; the first was composed of individuals from Jalapão, the second by individuals from Veadeiros, and the third was composed of individuals from Paranaíba and Canastra (Figure 6C).

The phylogenetic tree reconstructed with Bayesian Inference (Figure 7) reveals two large clusters, one composed of individuals from Paranaíba and Canastra and another composed of individuals from Veadeiros and Jalapão. More external to these two large groups, we have a group composed of three individuals from Canastra. However, this tree should be considered with caution, as no outgroup was used.

The population pairwise FST analysis showed significant values for all comparisons (*p* < 0.05) and indicated that the population Jalapão is the most differentiated among them, and that Canastra and Paranaíba populations are less differentiated between them (Table 1).

To perform the molecular analysis of variance (AMOVA), we tested several population groupings, using either all sampled individuals or excluding closely related individuals. The first grouping, as determined by Geneland, included populations of Canastra and Paranaíba in a single group, and Jalapão and Veadeiros were treated as separate populations. The second grouping followed the Structure result, where populations Jalapão and Veadeiros formed one group, while Canastra and Paranaíba formed another. A third grouping was divided into a group formed by Veadeiros and Jalapão populations, while Canastra and Paranaíba were treated as independent groups. The results indicated that most of the genetic variation was observed within populations, and the first grouping (Canastra + Paranaíba × Veadeiros × Jalapão) is the most likely clustering of populations, presenting the highest FCT values (0.0962 and 0.0750) and the lowest values of FSC (0.0979 and 0.0627), both with all individuals included and without closely related individuals, respectively (Appendix A).

The Mantel test between all population localities did not present a significant result (*p* > 0.05), which indicates that the geographic distance alone is not able to explain the molecular variance.

## 4. Discussion

A high level of inbreeding was observed in most of the Brazilian Mergansers of the four remaining areas of occurrence (Figure 3). However, in a single locality (Paranaíba), we observed the two individuals with the highest (PAR206) and lowest (PAR180) inbreeding coefficients (F), which indicates an important inter-individual heterogeneity of the wild Brazilian Merganser population.

The close genetic relationship between individuals of the same localities demonstrated by the pairwise kinship analyses (Appendix A) is somewhat expected since the Brazilian Merganser is a non-migratory bird [13,20], and it is known to occur historically in low population densities and disjunct areas [6]. This pattern may also be related to a high dependence of the Brazilian Merganser to its territory with “undisturbed” fast-flowing clear rivers close to water springs [6,13,20].

The very close relationship (first degree) between four individuals (JAL202, JAL203, JAL204, and JAL205) from Jalapão is expected because they are full siblings (captive adults derived from eggs collected in the same nest in the Jalapão region). A first-degree relationship was also evidenced in four individuals (PAR206, PAR207, PAR208, and PAR209) from Paranaíba, who are also full siblings that are currently captive adults derived from four eggs collected in a nest monitored yearly in the Paranaíba region (unpublished information from PAN pato-mergulhão, Brazil).

The average kinship of third degree for the PAR180 individual with 16 of 30 individuals from different locations cannot be simply explained by large distance gene flow [13,20]. To date, little is known about its dispersal mode [2], but it is unlikely that an individual can migrate large distances such as between Canastra (Minas Gerais state) and Jalapão (Tocantins state), which are about 1000 km apart (Figure 1). However, PAR180 also has the lowest individual inbreeding coefficient (−0.32184), which may inflate the overall kinship relationships, because this type of analysis is based on the difference between shared heterozygosity and homozygosity [40]. It may have also influenced the kinship estimates of CAN018, which has an inbreeding coefficient of 0.31249 and presents a significant kinship (second degree) only with individual CAN148. However, given the small population size and low genetic diversity observed within the species, drift and inbreeding may partially explain these results.

The global view of kinship values between individuals presented in the heatmap (Figure 2) illustrates the high differentiation of Jalapão and Veadeiros individuals, as compared to Canastra and Paranaíba. In the heatmap, we observe a closer relationship between Paranaíba and Canastra individuals, a fact that can be explained by the geographic proximity of these areas. In our records, the two spatially closest individuals from both areas (Canastra and Paranaíba) were originally captured at about 50 km in a straight line. Considering that each Merganser pair has a foraging area of 5 to 12 km of the river [12,20], and a former study on the dispersion of the Brazilian Merganser in Canastra identified the displacement of an individual to an area located 25.3 km away from its birthplace [18], meaning connectivity between Canastra and Paranaíba is very likely. This greater genetic similarity between individuals of Canastra and Paranaíba was also evidenced in the analyses of PCA (Figure 4) and DAPC (Figure 5), corroborating previous genetic studies with mitochondrial and microsatellite markers [22,24], which found greater similarity between Canastra and Paranaíba populations when compared with Veadeiros.

The number of populations and their geographic boundaries inferred by Geneland (Figure 6C), which showed us that Canastra and Paranaíba populations were grouped into a single cluster, are also consistent with our previous analyses. This result agrees with the known ecology of the species, reaffirming the Merganser’s philopatry to its habitat and area of occurrence, as well as its non-migratory characteristic [6,13,18,20]. This pattern was also supported by Structure (Figure 6B) when we analyzed the possible population groups without geographic bias. However, unlike Geneland, Structure grouped Canastra and Jalapão populations into a single cluster. This difference found by Geneland may have been influenced by the geographic distance between the two populations and the small population size of Jalapão (four full siblings), which together with a low FST may have overestimated the population inference [58,59]. When we analyzed the probabilities of each individual belonging to a determined population estimated by Compoplot (Figure 6A), we identified the same pattern found in the previous analyses, showing Canastra and Paranaíba with great genetic similarity between them, and Veadeiros and Jalapão with a slight similarity between them but very different from Canastra and Paranaíba.

The phylogenetic tree (Figure 7) shows us the same pattern of population clustering that we found in previous analyses, composed of two major groups, Canastra + Paranaíba and Veadeiros + Jalapão, as indicated in other analyses. However, three Canastra individuals (CAN098, CAN099, and CAN100) formed an external cluster related to the two largest ones. These individuals are likely full siblings because of a high kinship coefficient (0.25) that was also confirmed by fieldwork data and, consequently, share many alleles, forming a separate cluster, although in an unrooted BI tree (Figure 7).

The negative values of inbreeding coefficients found in some individuals (Figure 3, F < 0), including Jalapão individuals JAL204 and JAL205 that make up the captive population, suggest an excess of heterozygosity when compared to the population average [60]. Further, these individuals are important founders to increase the genetic viability in the captive population in further generations.

Using pairwise FST between populations (Table 1), we observed that Canastra and Paranaíba presented small genetic differentiation (0.09613), indicating a closer relationship. The most differentiated population is Jalapão, presenting the highest pairwise FST values (0.19 to 0.25). The significant FST values between all Brazilian Merganser localities suggest that there is a moderate population structure, even though this result should be taken with care because of the sampling representation of Jalapão composed of full siblings.

The AMOVA results for the three evaluated population groupings indicated that most of the genetic variation is between individuals/within populations. The results with and without related individuals indicated that grouping 1 (Canastra + Paranaíba × Veadeiros × Jalapão) is the most likely hierarchical clustering of populations because it has the highest FCT values (0.0962 and 0.0750) and the lowest FSC values (0.0979 and 0.0627), respectively. The FST values for the three tested groupings were moderate and significant (between 0.1090 and 0.1847), indicating that there is some level of population structure likely related to restricted gene flow between the current remaining populations and/or extinction of intermediate populations.

The overall results indicate a moderate degree of population structure and low genetic variability within populations that are composed of closely related individuals. The Canastra and Paranaíba populations are genetically and geographically close, even though individuals from both areas are distributed in separate genetic clusters. The Jalapão and Veadeiros populations appear to be less related between them, as well as farther apart, when compared with Canastra and Paranaíba populations. Anyway, Jalapão and Veadeiros populations are the ones with apparently lower density (and census) of Mergansers and should be better sampled in the future to allow for more precise estimates. Finally, the moderate degree of population structure concerning the four remaining areas of occurrence indicates that Brazilian Mergansers from all sources should be used for in situ and ex situ management in future translocation and reintroduction strategies.

## 5. Conclusions

The population genomic analysis of the Brazilian Merganser revealed a moderate population structure, low genetic diversity, and high inbreeding in the four remaining populations of this critically endangered species. However, the results should be taken with caution because of sampling bias, particularly for Jalapão (four full siblings) and Veadeiros, who are represented by few individuals. New population genomic studies should include more unrelated individuals from all areas, analyzing the use of a reference genome of the species that is under preparation (FRS, personal communication).

The genetic results presented in this study indicate that urgent and effective actions for *Mergus octosetaceus* conservation should be taken to avoid population decline. For example, the mapping of areas where the species can potentially survive, reproduce, and/or occurred historically may reveal appropriate areas to receive individuals with adequate “genetic values” to start a pilot reintroduction project. A captivity program has been established for Brazilian Mergansers in Itatiba Zoopark (São Paulo state) for more than a decade, where more than 60 captive individuals are currently kept that were derived from founding individuals of all four known remaining localities. These ex situ individuals are being genetically monitored by our research group to indicate the best pairings of Mergansers to maximize genetic diversity and avoid inbreeding. Thus, the data generated in this study can help in the selection of individuals that may be used in reintroduction projects to repopulate new areas and/or to increase the genetic variability in current areas where there is a very low density of Brazilian Mergansers (for example, Jalapão).

## Figures and Tables

**Figure 1 animals-13-03759-f001:**
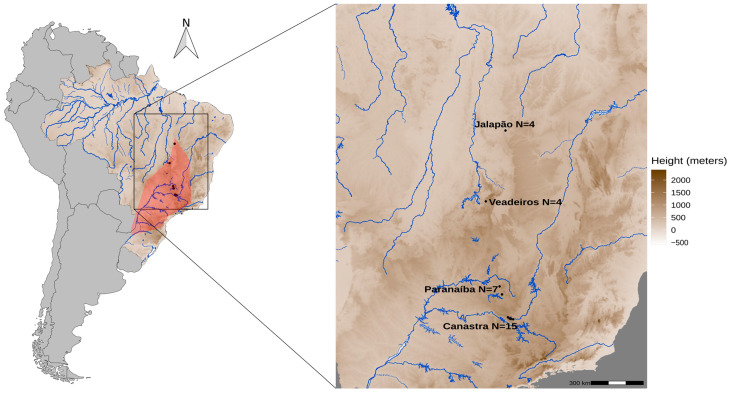
Historical and current population distribution of *Mergus octosetaceus*. The red area highlights the historical geographic occurrence of *M. octosetaceus* across Brazil, Argentina, and Paraguay (left map). The black dots represent the collection sites of samples from all four remaining populations in central Brazil (right map in detail): Canastra—Serra da Canastra National Park; Jalapão—Jalapão State Park; Veadeiros—Chapada dos Veadeiros National Park; Paranaíba—Alto Paranaíba region.

**Figure 2 animals-13-03759-f002:**
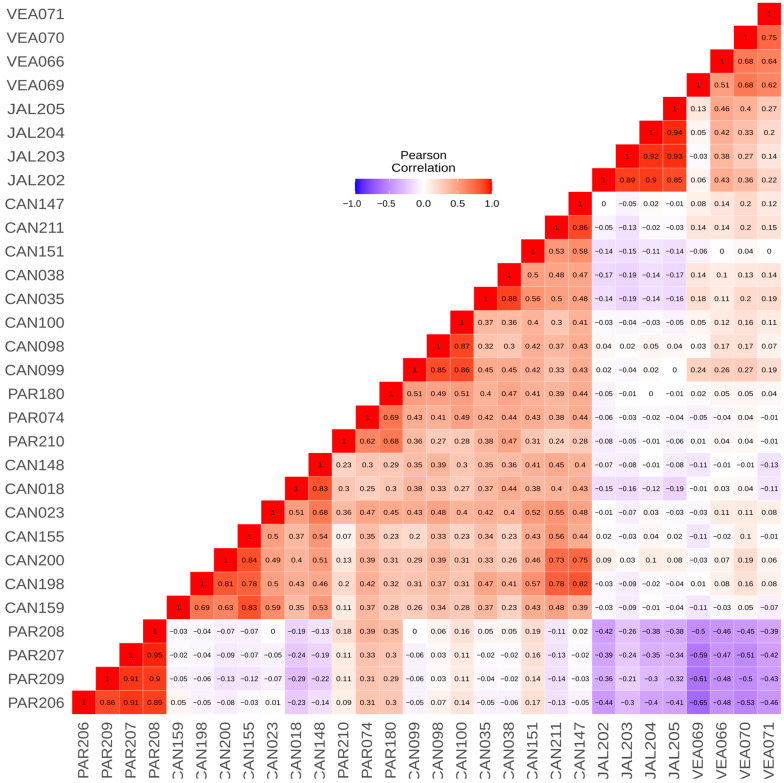
Heatmap depicting kinship indices, corrected by Pearson’s correlation matrix among individuals from Canastra (CAN). Paranaíba (PAR), Veadeiros (VEA) and Jalapão (JAL). The values on the map range from −1 (negative correlation, represented in blue) to 1 (positive correlation, represented in red).

**Figure 3 animals-13-03759-f003:**
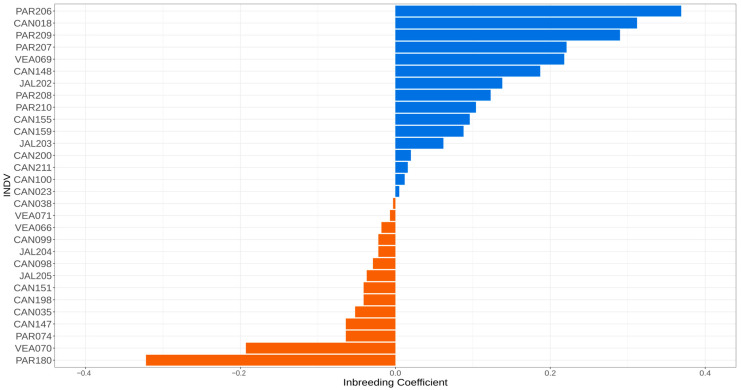
Inbreeding coefficients of 30 *Mergus octosetaceus* individuals from the four remaining populations (CAN—Canastra, PAR—Paranaíba, VEA—Veadeiros, and JAL—Jalapão). Positive inbreeding coefficients appear in blue (above) and negative ones in brown (below). The samples are in descending order of inbreeding coefficients.

**Figure 4 animals-13-03759-f004:**
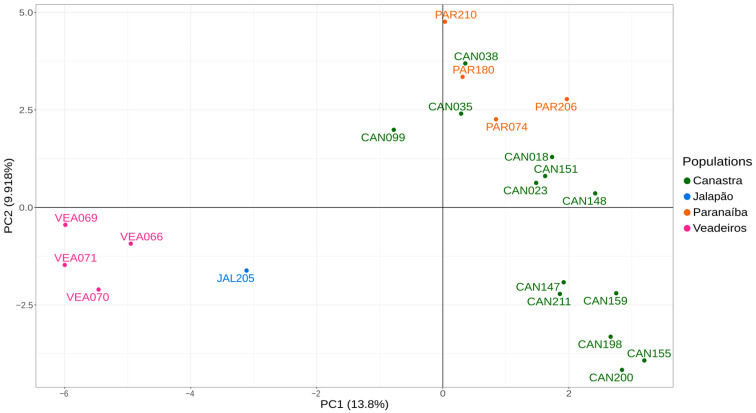
PCA analysis using 923 SNPs for 22 unrelated individuals. Green—individuals from Canastra (CAN); blue—individuals from the Jalapão (JAL); orange—individuals from the Paranaíba (PAR); pink—individuals from Veadeiros (VEA).

**Figure 5 animals-13-03759-f005:**
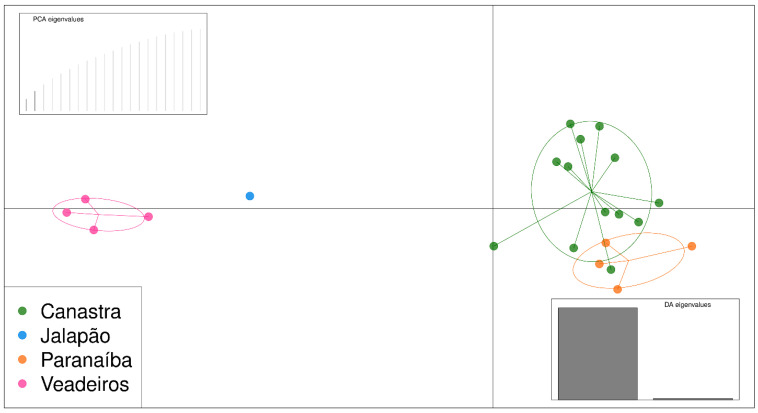
DAPC analysis using 923 SNPs for 22 unrelated individuals. Green—individuals from Canastra; blue—individuals from the Jalapão; orange—individuals from the Paranaíba; pink—individuals from Veadeiros.

**Figure 6 animals-13-03759-f006:**
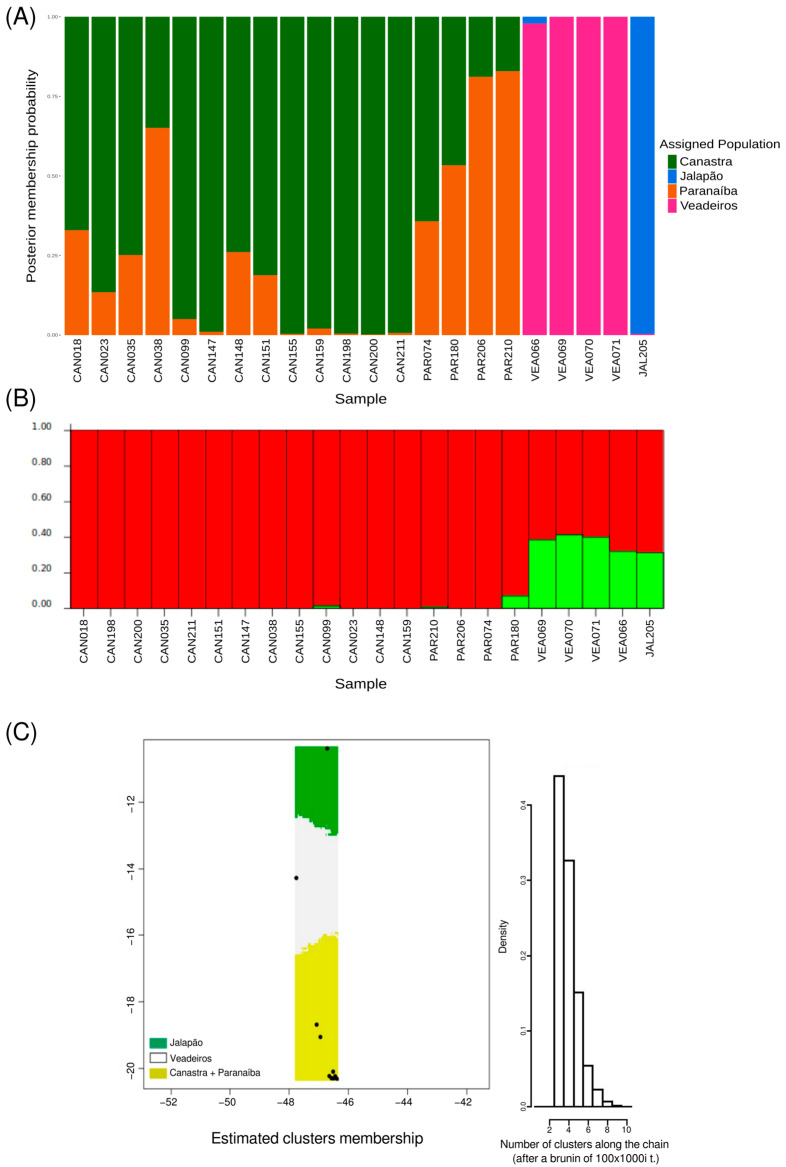
Population attribution analyses. (**A**) Graphic representation of the probability of every individual to belong to a determined “genetic population” (four assigned populations, k = 4) using adegenet/compoplot tool. (**B**) Structure result using the uncorrelated model and k = 2, excluding 1st-degree related individuals. Red—common alleles in the Canastra/Paranaíba cluster; Green—common alleles in the group formed by Veadeiros and Jalapão. (**C**) Geneland graphic showing the spatial distribution of three genetic clusters formed by *Mergus octosetaceus* individuals. Green—population of the Jalapão; White—population of Veadeiros; Yellow—populations of Canastra and Paranaíba. Geographic coordinates for samples/localities are shown as black dots (see Appendix A), and at right it is shown a graphic with the associated probabilities to the estimated number of clusters.

**Figure 7 animals-13-03759-f007:**
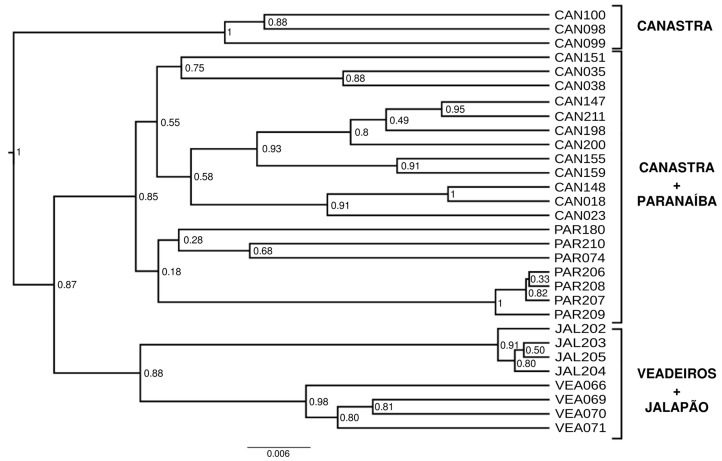
Phylogenetic tree (maximum clade credibility tree) using a Bayesian Inference with 923 SNPs from *Mergus octosetaceus* individuals. Posterior probabilities (PP) are depicted on each node. There is no outgroup assigned.

**Table 1 animals-13-03759-t001:** Pairwise FST analysis of *Mergus octosetaceus* populations defined a priori based on geographic locality. All pairwise FST values were significant (*p* < 0.05).

	Canastra	Paranaíba	Veadeiros	Jalapão
Canastra	0.00000			
Paranaíba	0.09613	0.00000		
Veadeiros	0.13071	0.20049	0.00000	
Jalapão	0.20018	0.25022	0.19839	0.00000

## Data Availability

The raw data file in VCF format for SNPs is available at the following DOI link: https://doi.org/10.5281/zenodo.10253118.

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
