# Peer review of "Population Genomics of the Critically Endangered Brazilian Merganser"

_animals, 2023, doi:10.3390/ani13243759_

Round 1

Reviewer 1 Report

Comments and Suggestions for Authors

The authors present an interesting study on population genomics of Critically Endangered Brazilian merganserMergus octosetaceus). Study design and research questions are clearly described. In this sense, it is easy to understand the aim of this study. The bright side of the manuscript is that to provide important results on the current population status and genetics of species using genomic markers. In this context, the study contributes to conservation and management of a Critically Endangered species. Only minor concerns are raised. Therefore, I would like to make some suggestions to improve the quality of the paper as below:

Line 28: “The number of threatened birds increased by about 30% in the last twenty years” -> “The number of threatened birds worldwide increased by about 30% in the last twenty years”

Lines 34-35: “The Brazilian merganser, Mergus octosetaceus, Vieillot, 1817 is the only Mergini duck of the Anseriformes order that is found in the southern hemisphere” Please add a reference.

Line 38-40: “Although population size studies are some-what speculative, it is estimated that there are less than 250 mature individuals in the wild and a decreasing population trend” Please rephase the sentence. For instance, “Although population size of the species is not clear, it is known that less than 250 mature individuals live in the wild and it is population trend is decreasing”.

Lines 206-207: “The Stacks pipeline was performed using a de novo alignment approach, as Mergus octosetaceus has no available reference genome.” Please add a reference for de novo alignment approach. For example, https://doi.org/10.1186/s13059-019-1774-4 and doi.org/10.1111/2041-210X.12775

Line 321: The Discussion section should be enriched with a more theoretical interpretation and relating the present results with additional concepts. For instance, the study results can be discussed in the framework of local genetic diversity and gene flow, inbreeding depression in the endangered species and subpopulation difference in different spices from different countries in broader context.

Line 419: The limitations of the study should be given in the conclusion section.

Line 453-454: Please change reference with current version:

“BirdLife International. 2016. Mergus octosetaceus. "The IUCN Red List of Threatened Species 2016: e.T22680482A92863947. en. Accessed 453on 09 April 2017. http://dx.doi.org/10.2305/IUCN.UK.2016-3.RLTS.T22680482A92863947.en.”

->

BirdLife International. 2019. Mergus octosetaceus. The IUCN Red List of Threatened Species 2019: e.T22680482A143756439.  https://dx.doi.org/10.2305/IUCN.UK.2019-3.RLTS.T22680482A143756439.en. Accessed on 07 November 2023.

Author Response

Thank you for all the suggestions that were used to improve our manuscript. For the de novo Stacks assembly we cited reference 35, which contains the protocol that we have used. We have also changed parts of Discussion and Conclusion to focus on the presented results and conservation meaning.

Furthermore, we have made a complete review of the text, figures, tables, and supplementary material.

Reviewer 2 Report

Comments and Suggestions for Authors

The article is well-presented and is suitable for publication in Animals with just a few minor changes or modifications. It's evident that the two populations of Mergus octosetaseus are distinct. However, I am curious which of the two populations is older, the Canastra + Paranaíba or the Jalapão + Veadeiros? Can we test it with the generated molecular data? Have any morphological or behavioral differences been observed between these two populations? Based on the current investigation, can we designate these populations as distinct subspecies?

The representation of the geographical and populational distribution of Mergus octosetaseus in Brazil in Figure 1 is informative. However, as the author mentioned that the historical distribution of the species encompassed three countries - Brazil, Argentina, and Paraguay, it would be more comprehensive if they could illustrate both the historical and present distribution.

It would be beneficial if the authors mentioned in the Materials and Methods section how many samples were collected from the historical sites and how many are from the current known distribution.

I'm also curious why no outgroup was used in the Bayesian analysis. It would be helpful if the authors could provide an explanation for this choice.

Adding the latitude and longitude of the sampling sites to Table S1 in the Supporting Information would be valuable for readers to understand the study's context better.

Finally, for Figure 2 and Figure 3, it would be ideal if the captions could be revised to make them self-explanatory.

Author Response

Thank you very much for your careful review of our manuscript. Because we have not used an outgroup, we have only depicted an "unrooted" Bayesian tree to illustrate the phylogenetic clustering of individuals in comparison to other clustering results (PCA, structure etc). We decided not to use a population tree at this moment because of the low sampling of some areas, e.g., Jalapão is represented only by four full siblings, thus not representative of the population whose census has been recently estimated to only 7 individuals (PAN - ICMBio 2023, unpublished). There are no morphological or behavioral differences up to date described for different populations, and we have found non-significant differences to claim them to be different subspecies. 

We have changed Figure 1 accordingly, thank you for pointing out this. We have added the historical distribution in Brazil, Paraguay and Argentina, and the four remaining populations in central Brazil (Cerrado biome).

All samples were from all four current populations in Brazil. We have no samples from the historical distribution, i.e., museum samples. Even though in the past we have PCR-sequenced 200 base pairs of mtDNA of two samples from Argentina (museum specimens), it is impossible to build a GBS or ddRAD library with this material. Indeed, we discarded some samples of the analysis because of low-quality data due to poor DNA quality. 

We have not used an outgroup in the Bayesian tree because our SNPs were identified with a de novo method in Stacks, we have no reference genome, as the use of the closest species with a good quality genome (Anas platyrhynchos) generated poor results, as both species are separated for many millions of years. Thus it was difficult to map most of SNPs in a reference genome, and it is why we are currently doing a high-quality genome for Mergus octosetaceus, which will allow a fine mapping of the reads/SNPs and the use of outgroups.

We added the geographic coordinates to Table S1, and revised the captions of all figures, as requested.